# Nutritional status, dementia, and mobility among nursing home's residents: First exhaustive cross-sectional study in Limousin territory (France)

**Philippe Fayemendy**[1,2,3], **Gustave Mabiama**[1,2,4]*, **Thibault Vernier**[1], **Aude Massoulard-Gainant**[3,5‡], **Carole Villemonteix**[3‡], **Jean-Claude Desport**[1,2,3], **Pierre Jésus**[1,2,3]

1 Nutrition Unit and Specialized Centre for Obesity of the Limousin territory, University Hospital, Limoges, France, 2 Inserm U1094, Limoges University, IRD, Tropical Neuroepidemiology, Epidemiology and Neurology Tropical Institute, GEIST, Limoges, France, 3 Health Regional Agency Network Limousin Nutrition (LINUT), Isle, France, 4 Microbiology, Immunology-Hematology and Morphologic Sciences Laboratory (LMIHSM), Doctoral Training Unit in Health Sciences (UFD-SCS), Doctoral School, Douala University, Douala, Cameroun, 5 Home Hospitalization and Geriatric Service, University Hospital, Limoges, France

☙ These authors contributed equally to this work.
‡ These authors also contributed equally to this work.
* gustave.mabiama@unilim.fr

**Data Availability Statement:** Under the agreements signed between Elderly Nursing homes and the Limousin Nutrition network

## Abstract

### Background

Aging is accompanied by a drop in the level of health and autonomy, within Western countries more and more people being cared for in nursing homes (NH). The nutritional data in NH in France remain poor, not exhaustive and not representative. The objective of the study was to assess the nutritional status, dementia and mobility patterns among residents of NH in the Limousin territory of France.

### Methods

The study was cross-sectional, descriptive and exhaustive, conducted with the residents of 13 voluntary NH. Undernutrition was identified using French High Authority for Health criteria, and obesity if Body Mass Index >30, in the absence undernutrition criterion. The Mini Mental State examination scores was used for dementia assessment at the threshold of 24. The Mini Nutritional Assessment™ was used for mobility assessment. The statistics were significant at the 5% threshold.

### Results

866 residents (70.6% women) included with an average age of 85.3 ± 9.3 years. Undernutrition was 27.5%, obesity 22.9%, dementia 45.7% and very low mobility 68.9%. Women were older than men, more often undernourished, more often demented and more often had very low mobility (p<0.01). Undernutrition (p<0.0001) and low mobility (p<0.0001) were significantly higher among those with dementia versus those without dementia. Very low mobility was higher among undernourished (p<0.05).

(LINUT), health network of the Nouvelle Aquitaine Regional Health Agency, data are collected, saved, and protected within the LINUT network. These data are linked with personal information of each resident followed by the network, and are not accessible, because considered as medical files. These third-party data are linked with personal information of each resident followed by the network, and are not accessible, because considered as medical files. The restrictions were imposed by LINUT'S network. Any request can be addressed to Dr Jean Louis Fraysse, 16 Rue du Cluzeau, 87170 Isle, frayssejl@limut.fr.

**Funding:** The author(s) received no specific funding for this work.

**Competing interests:** No authors have competing interests

## Conclusions

Undernutrition and obesity are important problems in NH in France. Being a woman, having dementia and having a very low mobility may induce undernutrition.

## Introduction

Between 2008 and 2016, the aging of the population and the increasing dependency of the oldest people in France led to an increase of 2.7% per year in the number of NH places, which in 2017 reached 605,000 [1]. Nutritional problems are notable in these establishments, both in terms of undernutrition and obesity.

Undernutrition in the elderly has multiple consequences in terms of morbidity (increased risk of infection, fall, pressure ulcer, bone embrittlement, deterioration of the psychological state, reduction of autonomy and quality of life, etc.), and constitutes a risk factor for mortality in NH [2–4]. The prevalence of undernutrition is evaluated in France up to 10% for the elderly at home and up to 40% for residents of nursing homes [4]. At the international level, the range in nursing homes is wider (1.5–66.5%), depending on the populations studied and the tools used, with an average estimated at 20%.

Although obesity in the elderly has sometimes been considered favorable from the point of view of mortality or morbidity (obesity paradox) [5], for many pathologies moderate or severe obesity seems deleterious, because of an increase in the risk of falls, arthritis, diabetes, bedsores, a reduction in autonomy [6–9]. Its prevalence in French NH is not known, but French data in the general population show that the prevalence of obesity increases with age, at least up to 75 years [10]. Data in the United States indicates an increase in the prevalence of obesity in NH between 1992 (15%) and 2015 (27.9%) [9].

Both undernutrition and obesity are associated in NH with an increase in the demand for care, therefore at increased personnel and material costs, and this increase is particularly marked among obese residents [8, 11].

Furthermore, the link between dementia and undernutrition or the risk of undernutrition is well known [12–15], as well as the link between low mobility and obesity [12] or undernutrition and reduced physical capacity [16].

The nutritional data in nursing homes are very disparate with sometimes a lack of completeness and / or representativeness, whether they come from one or more nursing homes (Table 1) [2, 3, 9, 15, 16], and the prevalence figures specifically for France are based on evaluations or expert consensus than confirmed figures (Table 1) [17, 18].

The Limousin Nutrition network (LINUT) is a structure of the Regional Health Agency (ARS) of the New Aquitaine Region in France. Since 2004, this network has been providing nutritional care in nursing homes, and has a preventive role in these establishments [19].

The objective of the study was to analyze the data of exhaustive nutritional evaluation surveys carried out between 2008 and 2014 in NH in three departments of the New Aquitaine region, in connection with the presence of dementia and with the mobility of the residents.

## Materials and methods

### Ethical approval and consent to participate

The study was approved by the Ethics Committee of the Regional Health Agency Network (LINUT) prior to data collection. The data were collected within the framework of the agreements signed between the Limousin Nutrition network (LINUT), health network of the

**Table 1. Undernutrition and obesity among elderly in nursing homes in certain high-income countries.**

| First author / year | Country | n | Age (ys) | Women (%) | Undernutrition tools and thresholds | Undernutrition (%) | Obesity tools and thresholds | Obesity (%) |
|---|---|---|---|---|---|---|---|---|
| Challa / 2007** [2] | USA | 128 514 | | 74.4 | BMI<18.5 | 12.1 | | |
| Chan / 2010 [3] | Singapore | 154 | 77 | 53.2 | BMI<18.5 | 52 | | |
| | | | | | MNA<17 | 39 | | |
| Törmä / 2013 [16] | Sweden | 172 | 86.3 | 69.8 | MNA-SF | 30 | | |
| | | | | | BMI<22 | 40.7 | | |
| Vandewoude / 2019 [15] | Belgium | 2480 | 86.3 | 78 | MNA SF | 13.5 | | |
| Zhang / 2019**** [9] | USA | 1 743 443 (2005) | | | BMI<18.5 | 8.5 | BMI> = 30 | 22.4 (2005) |
| | | 1 517 872 (2015) | | | BMI<18.5 | 7.2 | BMI> = 30 | 27.9 (2015) |
| Jésus / 2012 [17] | France | 346 | 87.9 | 83.4 | BMI<24 or MNA™<17 | 53.5 | BMI > = 29 | 27.4 |
| Desbordes / 2018* [18] | France | 248 | 88 | 78.2 | BMI<21 or Alb<35 g/L or weight loss> = 5% 1 mo or > = 10% 6 mo | 39.5 | | |

MNA™: Mini Nutritional Assessment™; SF: Short Form; BMI: Body Mass Index; n: number; mo: month

*newly admitted residents

**representative data

****long stay residents (> = 100d/y).

Nouvelle Aquitaine Regional Health Agency and the Elderly Nursing Homes, which stipulate that patient data can be used for research purposes after approval by the patients or their legal representatives. The NH were all volunteers, they were contracted for clinical research with the LINUT network, and residents or their legal representatives had given their verbal informed consent for the assessments, noted on each residents' file by the LINUT dieticians.

## Consent for publication

According to the agreement signed between the Limousin Nutrition network (LINUT), health network of the Nouvelle Aquitaine Regional Health Agency and the Elderly Nursing Homes, data collected with the approval of patients or their legal representatives can be used for publication.

The exhaustive descriptive cross-sectional study covered 13 NH in the Limousin territory (including three departments and around 800,000 inhabitants), between 2008 and 2014. All residents of each NH were included, except those whose state of health, according to the doctor, did not allow the measurements or those who refused the measurements after being informed. The assessments were carried out by a doctor and a dietician from the network specializing in geriatrics, and lasted two to three days by NH, depending on the number of residents to be assessed.

Residents' weight was measured with underwear using an electronic scale (SECA 813, Hamburg, Germany, to the nearest 0.1 kg) or an electronic weighing pan (SECA 634, Hamburg, Germany, to the nearest 0.1 kg or A&Z Gastineau, Roissy, France, to the nearest 0.1 kg). The size was measured using a wall measurement to the nearest 0.1 cm. For patients who cannot be verticalized, measuring the heel-to-knee distance using an appropriate measuring rod (Securimed 2068, Cappelle-la-Grande, France, to within 0.5 cm) made it possible to obtain the size using the Chumlea formulas [20]. The nutritional status of residents was determined according to the French criteria of the Haute Autorité de Santé (HAS) of 2007 for people over the age of 70 [21], and the 2003 French criteria for people under the age of 70 [22].

Undernutrition was present if i) the BMI was <21 for residents over the age of 70, and <18.5 for other residents, or if ii) the weight loss was > = 5% in one month or > = 10% in six months, or else iii) the Mini Nutritional Assessment (MNA™) was <17 in residents over 70 years of age. Overweight was defined by the existence of a BMI between 25 and 29.9 for people under 70, or between 27 and 29.9 for people over 70 years old. Obesity was defined by the existence of a BMI > = 30, whatever the age. For cost reasons and possible differences in normality limits depending on the measurement laboratory, albuminemia was not achieved.

The Mini Mental State (MMS) was used to determine whether or not dementia was present at the threshold of 24 [23]. Mobility was assessed using an MNA™ criterion, according to two classes: "very low mobility", corresponding to a bedridden state or the resident's ability to move only from bed to chair, "poor or good mobility" corresponding mobility possible inside and outside the NH. This MNA™ criterion was applied regardless of the age of the residents. The study was validated by the Ethics Committee and the Board of Directors of the LINUT Network, a structure of the Agence Régionale de Santé Nouvelle Aquitaine, in accordance with the agreement signed with the nursing homes for the use of data collected for scientific purposes, after acceptance by the residents or their legal representatives.

The databases were set up prospectively. They were processed anonymously using Statview™ software (SAS, Cary, USA). The quantitative values are presented as means ± standard deviation and the qualitative values as percentages. The analysis used Fischer's exact test, Student's t test and ANOVA. There was no multivariate analysis, given the limited number of criteria available in NH. The significance threshold was set at p<0.05.

## Results

The assessments covered 866 of the 1,043 residents of NHs surveyed (83%), or 66.6±14.7 residents/NH, and 17.0% of the residents could not be assessed either because of their refusal or because of their fragility. The M/F ratio was 0.42, or 70.6% of residents are women. The average age of residents was 85.3±9.3 years, the average BMI 26.4±5.3. Dementia was identified in 45.7% of the cases, and 31.1% of the residents were bedridden or could only go from bed to chair. Undernutrition affected 27.5% of residents, obesity 22.9% (Fig 1).

Women were older than men (87.1±8.1 versus 80.7±10.4 years, p<0.0001), more often undernourished than men (30.7% versus 20.0%, p = 0.01), with a tendency to be overweight less often (15.0 versus 20.4%, p = 0.058) (Table 2). They were more often demented (51.9% versus 30.8%, p<0.0001) and more often had very low mobility (66.3% versus 75.4%, p = 0.02).

The nutritional status was different (p<0.0001) depending on the presence or the absence of dementia (Table 2). Compared to non-demented residents, the demented group were more often undernourished (p<0.0001), less often overweight (p = 0.01) or obese (p = 0.004) and more often very limited in terms of mobility (38.9% of cases versus 23.9%, p<0.0001).

The nutritional status was different (p<0.0001) depending on mobility (Table 2). Residents with very low mobility were more often undernourished (p<0.0001) less often with normal nutritional status (p = 0.01) or overweight (p<0.0001) than those with poor or good mobility. There was no significant difference for obesity.

There was no significant difference for nutritional status according to NH, either in overall analysis of the four classes of nutritional status (p = 0.62) or by comparing the class of under-nourished residents with the others (p = 0.60) or by comparing the class of obese residents to the non-obese (p = 0.55). There was no significant difference for the nutritional status according to the year of the assessment, whether in overall analysis of the four nutritional status classes (p = 0.97) or by comparing the class of undernourished residents to that of non-undernourished (p = 0.85) or by comparing the class of obese residents to that of non-obese (p = 0.13).

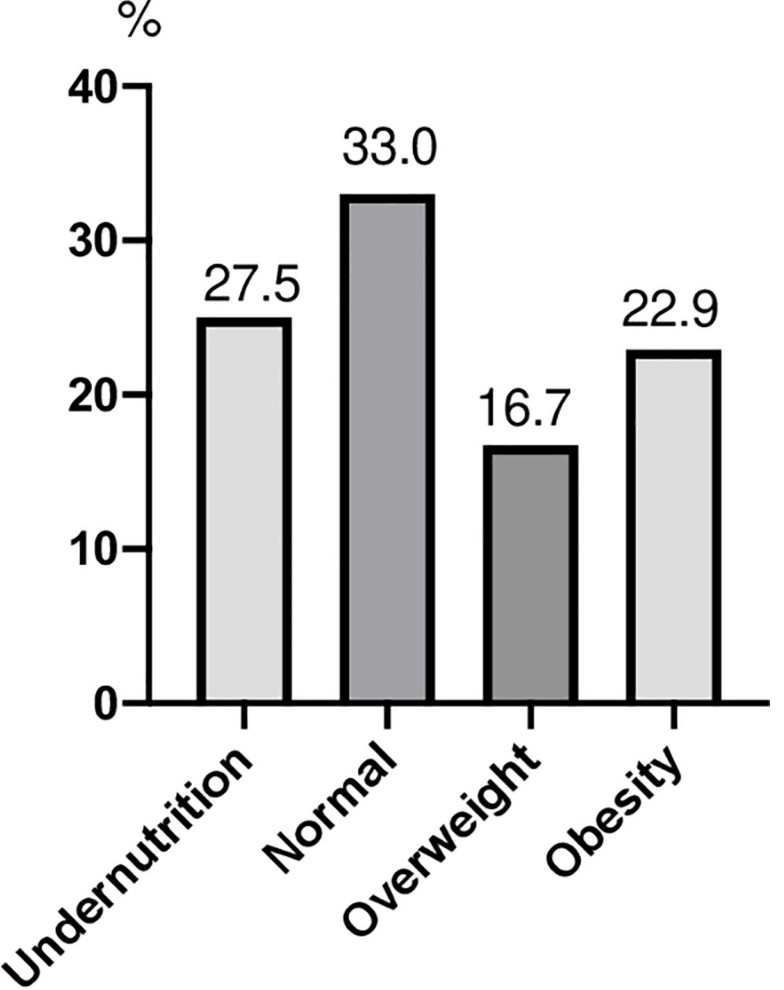

**Fig 1. Nutritional status of all residents assessed.**

## Discussion

This study is the first in France which gives exhaustive results on the nutritional status of residents in NH. It is not possible to claim that it is representative of all French nursing homes, but it nevertheless seems robust. Indeed, the average age of residents was close to that of all French NH (85.3 and 86.4 years respectively) [24] and the same was true for the sex ratio

**Table 2. Nutritional Status of residents based on gender, dementia, and mobility.**

| Nutritional status | Gender | | | Dementia | | | Mobility | | |
|---|---|---|---|---|---|---|---|---|---|
| | Women (%) | Men (%) | p | With (%) | Without (%) | p | Very low (%) | Poor or good (%) | p |
| | | | <0.0001 | | | <0.0001 | | | <0.0001 |
| Undernutrition | 30.7 | 20.0 | **0.01** | 34.4 | 19.2 | **<0.0001** | 49.5 | 16.7 | **<0.0001** |
| Normality | 32.2 | 34.9 | >0.05 | 32.2 | 32.2 | >0.05 | 22.7 | 32.2 | **0.01** |
| Overweight | 15.0 | 20.4 | 0.058 | 12.9 | 19.9 | **0.01** | 7.9 | 20.5 | **<0.0001** |
| Obesity | 22.1 | 24.7 | >0.05 | 19.5 | 28.5 | **0.004** | 19.9 | 25.5 | >0.05 |

Statistical tests used: ANOVA (or Kruskal-Wallis); Bold values indicate p<0.05.

(respectively 70.6 and 73.6% of women) [24] and the prevalence of dementia (45.7 and 40.0% respectively) [24]. Finally, the study covered all residents of the establishments investigated, and this exhaustiveness was only found in one French study, but which related to a single nursing home [18].

In our study, the prevalence of undernutrition was probably underestimated, due to the fact that albuminemia was not taken into account. This prevalence (27.5%) corresponds to the estimate in France, which is 14.5 to 60.8% [17, 18] (Table 1). In a previous French study [17], the threshold for BMI was 24, and undernutrition was samely overestimated. The difference with the prevalence of undernutrition in the 2012 French survey in the general population is major, since this figure was 2.6% for those aged 80 and over [10], but this survey focused on elderly people at home, a priori in better health than people in NH. Undernutrition is therefore an important problem in NH in France.

At an international level, the range of prevalence is even wider (1.5–71%) (Table 1), probably linked to very different populations and the use of equally different tools and thresholds for defining undernutrition. One recent study carried out in the USA have found a prevalence of less than 10% [9], the criterion was the existence of a BMI<18.5, with high prevalence of obesity as a counterpoint.

The higher prevalence of undernutrition in women appeared to be related to older age and the higher frequency in women with dementia and very low mobility. The increase in the risk of undernutrition in women that we have found is not always reported [2, 13, 25]. Given the predominance of women in nursing homes, and the frequency of co-morbidities in them, we can think that additional studies could be interesting on this subject.

There are many explanations for the link between dementia and undernutrition. Indeed, there may exist in demented residents, anorexia, food refusal, difficulty in eating related to disorders of recognition of food or cutlery, an increase in the energy expenditure of physical activity in the wandering dementia [12, 26, 27]. Our study showed that demented residents had lower mobility than non-demented people, and yet were less often overweight, which suggests, for the population studied, that food was insufficient. Overall, the study seems to indicate that, for demented residents, good availability of staff, careful nutritional monitoring and early management of undernutrition are necessary as already pointed out [26, 28, 29]. Regarding the association between very low mobility and the presence of undernutrition, our results are in agreement with other studies [28, 29]. Of course, the relationship between the two criteria can be understood in both directions: undernutrition can lead to sarcopenia, with a reduction in mobility, and low mobility can limit autonomy and the possibilities of eating. The higher prevalence of dementia may be a major reason, but other intercurrent conditions conventionally providing undernutrition (infections, pain, depression, neurological diseases, etc.) can lead to a decrease in mobility and therefore interfere.

The prevalence of obesity among the residents we studied was 22.9%. This result seems robust, because patients with BMI in an obesity zone, but who had an involuntary weight loss were not counted as obese. To our knowledge, there is only one French study on obesity in NH, which is neither representative nor exhaustive [17], and which reports 27.4% of obese residents in a selected population of 346 residents. Recent international results, mainly from the USA, indicate a prevalence range of 15 to 30.7% (Table 1). In addition, according to the 2012 French survey of the general population, 16% of people aged 80 and over were obese [10]. These results show that obesity is probably more frequent in nursing homes than in the general population of elderly people and therefore should not be underestimated. Indeed, it can increase the risk of morbi-mortality in the elderly [5] and, as in non-elderly adults, it can cause complications that lead to a reduction in quality of life and overuse of drugs [8]. The increase in the prevalence of obesity in nursing homes observed in the USA [9, 11] can obviously affect

European countries, with important consequences in terms of costs, linked to the increase in needs in personnel and adapted materials [9, 11]. This should be the subject of reflection within health systems.

The absence of differences in the nutritional status of residents over the seven-year period considered suggests that the problems do not seem to vary over time in the region studied. For obesity, in particular, the situation seems stable, in contrast to the data in the USA [9]. It would however be interesting to repeat this study annually on a longer period.

The study has several limitations. In fact, 17.0% of residents could not be assessed either because of their refusal, or because of their very fragile condition. It was therefore not possible to compare residents excluded from included. However, the excluded were probably seriously handicapped, since the NH where the exclusions were the most numerous (n = 45) was also the only one that received almost only dementias. There is therefore a good chance that exclusions, combined with the impossibility of obtaining an albuminemia have reduced the prevalence of undernutrition. Likewise, the prevalence of dementia and poor physical activity were likely to be reduced. It is more likely that finer criteria could have been used, in particular regarding patient mobility. An ADL score [30] would have given more precise elements. Regarding dementias, the seriousness of the disease has not been recorded. These details could not be obtained for reasons of availability of the evaluation staff, or else, if these elements were present in the residents' files, their time lag made them unusable. Finally, for reasons of team availability and access to files, the study was unable to integrate several factors of clinical condition as well as biological data from residents in NH. This is the main reason why we did not carry out a multivariate analysis of the criteria linked to undernutrition or obesity.

## Conclusion

The nutritional screening carried out in 13 NH with 866 residents revealed undernutrition in 27.5% of cases and obesity in 22.9% cases, in the ranges of the usual values in NH. Undernutrition and obesity in NH are notable problems in France, compared to elderly people at home. The consequences in terms of staff workload and costs are probably significant. The women were older, more often malnourished, demented and had more limited mobility than the men. Undernutrition and low mobility were significantly higher among those with dementia versus those without dementia. Very low mobility was higher among undernourished.

## Acknowledgments

We thank the LINUT network, caregivers, and residents of nursing homes of Limousin territory.

## Author Contributions

**Conceptualization:** Philippe Fayemendy, Jean-Claude Desport, Pierre Jésus.

**Data curation:** Aude Massoulard-Gainant, Jean-Claude Desport.

**Formal analysis:** Jean-Claude Desport.

**Investigation:** Philippe Fayemendy, Aude Massoulard-Gainant, Carole Villemonteix, Pierre Jésus.

**Methodology:** Jean-Claude Desport, Pierre Jésus.

**Supervision:** Jean-Claude Desport.

**Validation:** Jean-Claude Desport, Pierre Jésus.

**Visualization:** Gustave Mabiama.

**Writing – original draft:** Philippe Fayemendy, Gustave Mabiama, Thibault Vernier, Jean-Claude Desport, Pierre Jésus.

**Writing – review & editing:** Philippe Fayemendy, Gustave Mabiama, Jean-Claude Desport, Pierre Jésus.

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
