## [Decision Letter · Decision Letter 0]

21 Jan 2021

PONE-D-20-40393

Nutritional status, dementia and mobilityamongnursing home’s residents:First exhaustive cross-sectional study in Limousin Territory (France)

PLOS ONE

Dear Dr. Mabiama,

Thank you for submitting your manuscript to PLOS ONE. After careful consideration, we feel that it has merit but does not fully meet PLOS ONE’s publication criteria as it currently stands. Therefore, we invite you to submit a revised version of the manuscript that addresses the points raised during the review process.

I am very much thankful to the reviewers for their deep and thorough review.

Some minor revisions are required. Please carefully check the reviewers’ comments.

We look forward to receiving your revised manuscript.

Kind regards,

Sıdıka Bulduk, Prof. Dr.

Academic Editor

PLOS ONE

Journal Requirements:

2. Please amend your current ethics statement to clarify whether the IRB institute approved the study protocol prior to data collection.

Furthermore, Please provide additional details regarding participant consent. In the ethics statement in the Methods and online submission information, please ensure that you have specified (1) whether consent was suitably informed and (2) what type you obtained (for instance, written or verbal).

3. We noted in your submission details that a portion of your manuscript may have been presented or published elsewhere.

"Only abstract was published in Clinical Nutrition ESPEN, December 2020."

Please clarify whether this publication was peer-reviewed and formally published. If this work was previously peer-reviewed and published, in the cover letter please provide the reason that this work does not constitute dual publication and should be included in the current manuscript.

Reviewers' comments:

Reviewer's Responses to Questions

**Comments to the Author**

1. Is the manuscript technically sound, and do the data support the conclusions?

Reviewer #1: Yes

Reviewer #2: Yes

2. Has the statistical analysis been performed appropriately and rigorously? 

Reviewer #1: Yes

Reviewer #2: Yes

3. Have the authors made all data underlying the findings in their manuscript fully available?

Reviewer #1: Yes

Reviewer #2: Yes

4. Is the manuscript presented in an intelligible fashion and written in standard English?

Reviewer #1: Yes

Reviewer #2: Yes

5. Review Comments to the Author

Reviewer #1: ts of 13 voluntary NH. Undernutrition was identified using French High

Authority for Health criteria, and obesity if Body Mass Index >30, in the absence

undernutrition criterion. The Mini Mental State was used for dementia assessment at

the threshold of 24. The Mini Nutritional Assessment TM was used for mobilitity

assessment. Significant statistics at the 5% threshold. Article is a current topic and written in good language. It is suitable to be published in the journal

Reviewer #2: Review Comments to the Author

Please use the space provided to explain your answers to the questions above. You may also include additional comments for the author, including concerns about dual publication, research ethics, or publication ethics. (Please upload your review as an attachment if it exceeds 20,000 characters) (Limit 200 to 20000 Characters)

Please see my attached review, the manuscript can be published after corrections

6. PLOS authors have the option to publish the peer review history of their article (what does this mean?). If published, this will include your full peer review and any attached files.

Reviewer #1: **Yes: **Prof. Hande Sahin

Reviewer #2: No

---

## [Author Response · Author response to Decision Letter 0]

2 Apr 2021

My co-authors and I would like to thank you for reviewing, comments and suggestions made on our article. You will find in this letter the answers to the various points raised.

1. The study was approved by the Ethics Committee of the Regional Health Limousin Nutrition Network (LINUT) prior to data collection.

2. The Nursing Homes (NH) were all volunteers, they were contracted for clinical research with the LINUT network, and residents or their legal representatives gave their verbal informed consent for the assessments, noted on each residents’file by the LINUT dieticians.

3. Part of the work was presented by Pierre Jesus as e-poster (P419 - ASSESSMENT OF THE NUTRITIONAL STATUS OF 866 OLDER PEOPLE IN NURSING HOMES, AND LINKS WITH DEMENTIA AND MOBILITY) at the virtual congress on Clinical Nutrition and Metabolism organized by ESPEN from 19 to 21 September 2020. This e-poster published by Clinical Nutrition ESPEN 2020, 40, pp.590 does not value all aspects of the work done and results obtained, hence the idea of publishing the entire article in PLOS ONE.

4. The data are the property of LINUT network, a network of the Nouvelle Aquitaine Health Regional Agency. These data are linked with personal information of each resident followed by this network, and are not accessible, because they are considered as medical files. These third-party data are linked with personal information of each resident followed by the network, and are not accessible, because considered as medical files. 4. The data are the property of LINUT network, a network of the Nouvelle Aquitaine Health Regional Agency. These data are linked with personal information of each resident followed by this network, and are not accessible, because they are considered as medical files. The restrictions were imposed by LINUT'S network. Any request can be addressed to Dr Jean Louis Fraysse, 16 Rue du Cluzeau, 87170 Isle, frayssejl@limut.fr

Hoping you will agree with our responds. 

Sincerely.

Gustave MABIAMA.

---

## [Editor Report · Decision Letter 1]

12 Apr 2021

Nutritional status, dementia and mobility among nursing home’s residents : First exhaustive cross-sectional study in Limousin Territory (France)

PONE-D-20-40393R1

Dear Dr. Mabiama,

We’re pleased to inform you that your manuscript has been judged scientifically suitable for publication and will be formally accepted for publication once it meets all outstanding technical requirements.

Kind regards,

Sıdıka Bulduk, Prof. Dr.

Academic Editor

PLOS ONE

Additional Editor Comments (optional):

In the Abstract Section, please write within, not with in.

In the Highlights Section, the sentence should be written like that: Undernutrition and low mobility were significantly higher among demented persons.

Page 15, “walking dementia” ??. The authors should improve the sentence.
---

## [Editor Report · Acceptance letter]

19 Apr 2021

PONE-D-20-40393R1 

Nutritional status, dementia, and mobility among nursing home’s residents: First exhaustive cross-sectional study in Limousin Territory (France) 

Dear Dr. Mabiama:

I'm pleased to inform you that your manuscript has been deemed suitable for publication in PLOS ONE. Congratulations! Your manuscript is now with our production department. 

Kind regards, 

on behalf of

Dr. Sıdıka Bulduk 

Academic Editor

PLOS ONE